# Factors Associated with Lifestyle Habits and Mental Health Problems in Korean Adolescents: The Korea National Health and Nutrition Examination Survey 2017–2018

**DOI:** 10.3390/ijerph17249418

**Published:** 2020-12-15

**Authors:** Hana Yoo, Namhee Kim

**Affiliations:** 1Department of Nursing, Daejeon University, Daejeon 34520, Korea; hanayoo@dju.kr; 2College of Nursing, Yonsei University, Seoul 03722, Korea

**Keywords:** adolescent, health behavior, lifestyle, mental health problem, stress, depression, suicidal thoughts

## Abstract

To identify factors associated with Korean adolescents’ lifestyle habits and mental health problems (stress perception, depressive mood, and suicidal thoughts), data from 842 adolescents’ (13–18 years) responses to the Korea National Health and Nutrition Examination Survey (VII–2 and 3; 2017–2018) were analyzed. After controlling for sociodemographic and health status characteristics, results of multinomial logistic regression revealed drinking alcohol (stress perception: odds ratio (OR) = 1.61, 95% confidence interval (CI) = 1.19–2.19; depressive mood: OR = 2.59, 95% CI = 1.67–4.02; suicidal thoughts: OR = 2.28, 95% CI = 1.18–4.42), increased sedentary time (stress perception: OR = 1.88, 95% CI = 1.36–2.58), ≤6 h weekday sleep (stress perception: OR = 1.28, 95% CI = 1.00–1.66; suicidal thoughts: OR = 1.98, 95% CI = 1.05–3.75), and 8 h ≤ weekend sleep (stress perception: OR = 0.74, 95% CI = 0.56–0.98; depressive mood: OR = 0.63, 95% CI = 0.41–0.98; suicidal thoughts: OR = 0.41, 95% CI = 0.21–0.79) were significantly associated with mental health problems. Reducing Korean adolescents’ mental health issues may require alcohol education, timed breaks/reduced sedentary time, and campaigns encouraging adequate sleep for teenagers.

## 1. Introduction

Adolescence is a period in which individuals develop the lifestyle habits and attitudes that may influence their health in adulthood and learn behaviors for maintaining lifelong health [1]. Protecting adolescents from risk factors and adverse experiences that can affect their psychological well-being and growth potential is important for maintaining adolescents’ physical and mental health in adulthood [2]. Globally, 10–20% of adolescents experience mental health problems such as mood disorders, anxiety disorders, impulse control disorder, and substance use disorders [3]. Mental health problems account for approximately 16% of the disease and injury burden in teenagers [2,3]. Mental health problems experienced in adolescence cause various adverse outcomes and have a considerable impact on public health [2]. Therefore, it is important to recognize and pay attention to mental health problems in adolescents [2].

Globally, depression is one of the leading causes of disease and disability, and suicide is the third leading cause of death among adolescents aged 15–19 [2]. In the Organization for Economic Cooperation and Development countries, the prevalence of depression and suicide among adolescents in Korea is relatively high, and suicide is the number one cause of death in Korean adolescents [4,5]. Therefore, Korea considers adolescents’ stress, depression, and suicide as continuations of mental health problems, so they are considered important public health problems and are investigated each year through national surveys [6,7,8,9,10]. According to the 2019 Korea Youth Risk Behavior Web-based Survey (KYRBWS), Korean adolescents show a perceived stress rate of 39.9%, prevalence of depressive mood of 28.2%, and suicide attempt rate of 3.0% [7,8,9,10]. Furthermore, suicide has been the leading cause of death in the Korean adolescent population since 2007 [4], and the teenage suicide rate in Korea increased by 22.1% in 2018 from 2017, with mental difficulties being the reason for most of the suicides [5]. Considering these facts, it is clear that mental health problems are a major public health concern for Korean adolescents. 

Stress, depression, and suicide are on the mental health continuum and are correlated. First, adolescents experience various types of stress that potentially threaten their healthy development and well-being [11]. Accumulated stress or chronic stressors in adolescence lead to mental health outcomes such as depression, anxiety, suicide, and antisocial behaviors [11]. Although not all experiences of stress lead to depressive episodes, stress has been found to precede most cases of depression and diverse psychological problems, with stress and depression being correlated [11,12]. Second, depression in adolescence predicts various mental health problems and has been identified as the most important risk factor for suicide attempts and completed suicide in adolescents [13,14]. Third, the primary causes of suicide are depression and stressful life events [15]. Furthermore, suicide has been described as a continuum of thinking, beginning with suicidal ideation, which has been confirmed in adolescents as a strong emotional stressor and a predictor of suicide attempts, psychopathology, and poor functioning in adulthood [16]. Suicidal thoughts can lead to suicide attempts, and adolescents who experience suicidal thoughts have a 12-times higher risk of attempting suicide [16]. 

Adolescence is a period when people are vulnerable to exposure to health risk behaviors and the initiation of risk-taking behaviors [2]. Because adolescents’ mental health problems are clinically heterogeneous and the interaction of different health risk factors increases the risk of mental health problems, it is difficult to understand the etiology of mental health problems in adolescents [14]. Therefore, the World Health Organization (WHO) explains that alternatives to risk-taking behaviors should be strengthened as an intervention for mental health promotion and prevention in adolescents [2], and several studies have stated that reducing the emergence and exacerbation of health risk behaviors is the most important measure to promote adolescents’ mental health [17,18,19]. When considering the possibility of the co-occurrence of various health risk behaviors, it is necessary to consider various modifiable lifestyle habits to understand and manage the mental health problems of adolescents [18]. Therefore, it is necessary to identify modifiable lifestyle habits that are commonly associated with stress, depression, and suicidal thoughts on the adolescent mental health continuum.

Previous studies examined the association between various lifestyle habits and mental health problems in adolescents [2,20,21,22,23,24,25]. Smoking and drinking in adolescence have been shown to increase the probability of mental health problems such as stress, depression, suicidal ideation, and dissatisfaction with life in the short and long term [2,20,21]. Greater amounts of sedentary behavior were shown to increase depression and decrease satisfaction with life and happiness in adolescents [22], and physical activity was associated with lower levels of stress, depression, negative affect, and total psychological distress in adolescents [22,23]. Short sleep duration was shown to be correlated with a significantly higher risk for depression and suicidal ideation [24,25]. However, there are limited studies that have examined the association between mental health problems, sedentary time, and weekday and weekend sleep in Korean adolescents using large-scale national data from an anonymous survey. Technological advances and increased sedentary lifestyles in modern society have increased physical inactivity and sedentary behaviors, which, in turn, have led to elevated rates of mental illnesses and increased morbidity and mortality [26]. Adequate sleep duration is important for adolescents to be healthy because insufficient sleep in adolescents increases the incidence of stress, depression, and suicide [27,28]. Therefore, this study uses the variables of physical activity, sedentary time, and weekday and weekend sleep duration from a nationally representative sample of Korean adolescents recently generated by the 7th Korea National Health and Nutrition Examination Survey (KNHANES). In addition, this study explores the recent trends in lifestyle habits and mental health in the Korean adolescent population to enable the development of preventive measures to address modifiable lifestyle habits. An analysis of the data collected by the government may allow mental health care professionals to gain a comprehensive, integrated, and responsive understanding of adolescent mental health problems. The findings of these study could be generalized to Korean adolescents and it is expected to be the basis for policies, guidance development, and programs for promoting adolescents’ mental health.

This study aimed to examine mental health problems (stress perception, depressive mood, and suicidal thoughts) according to sociodemographic factors, health status, and lifestyle habits (smoking, drinking alcohol, sedentary time, physical activity, and sleep duration) and to identify the differences in the factors associated with mental health problems in a nationally representative Korean adolescent sample, specifically focusing on lifestyle habits. We hypothesized that (1) sociodemographic, health status, and lifestyle habit factors of adolescents would be different with regard to stress perception, depressive mood, and suicidal thoughts; and (2) smoking, drinking alcohol, long sedentary times, lack of physical activity, and short weekday and weekend sleep of adolescents would be associated with the likelihood of an increased stress perception, depressive mood, and suicidal thoughts of adolescents after controlling for sociodemographic and health status factors.

## 2. Materials and Methods 

### 2.1. Study Design

This study used the raw data from the seventh wave of the KNHANES (VII–2 and 3; 2017–2018) [6]. We employed a cross-sectional and descriptive correlational design to investigate the unique factors associated with mental health problems in Korean adolescents.

### 2.2. Description of the Primary Data and Participants

KNHANES is a nationwide health and nutrition survey [6]. KNHANES is conducted annually based on Article 16 of the National Health Promotion Act and consists of a health interview, health examination, and nutrition survey [6]. The Korea Centers for Disease Control and Prevention (KCDC) has managed KNHANES’ external quality control programs, internal quality assurance, and control procedures [29]. KNHANES data have been assessed for validity and reliability with regard to people’s health levels and health behaviors; these data represent health statistics that are used to establish national health plans and the National Health Plan 2020, and they have also been used in international comparative studies [6,29]. Because KNHANES is based on the most recent Population and Housing Census data available at the time of the sample design, it is possible to extract representative samples of all people over the age of one year in Korea [6]. The results of the seventh KNHANES, conducted from 2016 to 2018, were released in 2020 by the KCDC [6]. For the sample to accurately represent Koreans, the seventh KNHANES was based on the size of the geographical area and housing type to stratify the extraction frame: sex, age, proportion of living space, and education levels of household members were used for implicit stratification [6]. The KNHANES was conducted by a trained survey expert team at mobile examination centers and the participants’ homes. The present study used health interview and health examination data from the KNHANES VII–2 and 3 (2017–2018) [6]. The final sample of KNHANES VII–2 and 3 (2017–2018) was selected from 384 regions and 8832 households (192 regions and 4416 households per year) [6]. In total, 16,119 participants completed the KNHANES VII–2 and 3 (2017–2018) [6]. The final sample consisted of adolescents aged 13–18 years (*n* = 842) after excluding participants younger than 13 years and older than 18 years (*n* = 15,205) and those with missing data related to health status and lifestyle habits (*n* = 72; Figure 1).

### 2.3. Measurements

#### 2.3.1. Dependent Variables

Based on a completed self-report on questions selected by collecting opinions from the central government and experts of related fields, mental health problems (stress perception, depressive mood, and suicidal thoughts) were used as the dependent variables for this study. Stress perception was classified into two groups: high (I feel extremely stressed or I feel a lot of stress) and low (I feel a little stressed or I rarely feel stress) using the following question: “How much stress do you feel in your daily life?” [6]. Depressive mood refers to a continuous depressive mood for two weeks or more and it was also categorized into two groups, yes and no, using the following question: “Have you felt sad or desperate enough that it interfered with your daily life for more than two weeks in a row within the last year?” [6]. Suicidal thoughts were defined as thinking of suicide for one year and they were categorized into two groups, yes and no, using the following question: “Have you ever thought about committing suicide within the last year?” [6].

#### 2.3.2. Independent Variables

Sociodemographic, health status, and lifestyle habits characteristics were used as independent variables. Based on the completed self-reports, sociodemographic characteristics were investigated in terms of age, sex, residential area, and household income level. Participants’ ages were classified into two groups according to the age classification normative levels for middle- and high-school students in the Korean educational system [6,30]: 13–15 years old and 16–18 years old. Sex was categorized as either man or woman [6]. Residential areas were categorized as rural or urban [6]. Household income levels were surveyed for one adult per household (aged 19 and over) and categorized as low, middle-low, middle-high, and high, according to the reference income quartiles of the total household and total population [6].

Health status characteristics were investigated in terms of perceived health status and percentile of body mass index (BMI). Perceived health status was classified into three groups: poor (very poor, or poor), fair, and good (good, or very good) using the following question: “How would you rate your current health status?” [6]. Participants’ BMI percentile was used to categorize four weight-health groups: underweight (<5th percentile), normal (5th ≤ percentile < 85th), overweight (85th ≤ percentile < 95th), and obese (95th percentile≤) using the percentiles by gender and age given by the 2017 Korean Child·Youth Growth Chart [6,31].

Based on the completed self-report surveys, lifestyle habit characteristics were investigated in terms of smoking, drinking alcohol, sedentary time, physical activity, weekday sleep, and weekend sleep. Smoking was categorized as yes or no using the following question: “Have you ever smoked a cigarette, even one or two puffs?” [6]. Drinking alcohol was categorized as yes or no, using the following question: “Have you ever drunk more than one cup of alcohol in your life?” [6]. Sedentary time refers to the average amount of time per day sitting or lying down: sitting at a desk; sitting with a friend; traveling by car, bus, or train; reading books; writing; playing cards; watching TV; playing games (Nintendo, computer, PlayStation); using the Internet; listening to music; etc. [6]. Sedentary h were converted to sedentary minutes, which were then classified into three groups: ≤ 600 (≤25th percentile), 601–780 (25th < percentile ≤ 75th), and 781 ≤ (75th percentile<) min/day, using the following question: “Other than your usual sleeping time, how many hours per day do you sit or lie down while working, at home, when moving from place to place, or with friends?” [6]. Physical activity level, meaning the number of days a participant engages in physical activity for 60 minutes or more per day in a week, was determined using the followed question: “In the last 7 days, on how many days did your heart rate increase from normal or did you have a total of 60 minutes or more of physical activity per day (regardless of type) that left you out of breath?” [6]. The physical activity variable is only applicable to data for adolescents from KNHANES VII–2 and 3 (2017–2018) [6]. To focus on whether the participants engaged in physical activity, physical activity was classified as yes (1, 2, 3, 4, 5, 6, or 7 days/week) and no (no days/week). Weekday sleep and weekend sleep, which refer to the average nighttime sleep duration per day calculated in minutes between the time when participants went to bed and when they woke up, were determined using the following questions: “When do you usually go to bed and wake up on a weekday (or workday)?” and “When do you usually go to bed and wake up on the weekend (or the day you are not working, the day before you are not working)?” [6]. Weekday sleep and weekend sleep variables were newly created in KNHANES VII [6]. Based on the recommendations of the Korean Society of Sleep Medicine [32], weekday and weekend nighttime sleep were classified into three groups: ≤360, 361–480, and 481 ≤ min/day [6].

### 2.4. Statistical Analysis

KNHANES’ sample design was extracted using a two-stage stratified cluster sampling of survey regions and households [6]. To expand and interpret the results from the target population of the Korean people, a complex sample design was performed using the IBM SPSS Complex Samples version 25.0 (IBM, Armonk, NY, USA). The integrated weight of the two-year period (2017–2018) was calculated by multiplying the existing weight (2017–2018 association analysis weight of health interview and health examination) by the annual survey region ratio (192 regions annually/384 total regions for 2017–2018) [6]. This study analysis employed a two-tailed statistical level of significance of *p* = 0.05. In this study, non-responses were excluded from the analysis and imputation was not performed because missing values across all variables used were less than 0.8% and they consisted of system-missed values [6]. Using the data from KNHANES VII–2 and 3 (2017–2018), we analyzed 842 participants representative of Korean adolescents.

All analyses were conducted using a weighted sample of complex design. Participants’ characteristics were determined using descriptive statistics. Chi-squared tests were used to compare the differences in mental health problems’ data between sociodemographic, health status, and lifestyle habits characteristics. Multi-nominal logistic regression was performed to compare factors associated with mental health problems in Korean adolescents after controlling for their sociodemographic and health status characteristics (age, sex, residential area, household income level, perceived health status, and BMI percentile).

### 2.5. Ethics Approval

The 1st- and 2nd-year surveys of KNHANES were conducted without deliberation of the Institutional Review Board (IRB) in 2016 and 2017, because it corresponded to a study conducted directly by the state for the public’s welfare according to the Korean Bioethics and Safety Act. The 3rd-year survey of KNHANES was conducted with the approval of the IRB of the KCDC (IRB number: 2018-01-03-P-A). All study participants of KNHANES participated in the primary study after providing informed consent, and the KCDC provided only non-identifiable data according to the Personal Information Protection Act and Statistical Law. KNHANES data are publicly available for free and can be downloaded from the KCDC website (https://knhanes.cdc.go.kr) [6]. Therefore, this study did not require approval from the IRB.

## 3. Results

### 3.1. General Characteristics of the Participants

Table 1 shows the general characteristics of the participants. The study population consisted of 51.9% male and 48.1% female participants. Regarding health status, a total of 58.1% of the participants rated themselves as healthy and more male adolescents rated themselves healthy than did female adolescents. Regarding BMI, higher rates of both obesity and underweight were reported in male adolescents.

Regarding lifestyle habits, in male adolescents, the smoking rate and drinking alcohol rate were 18.7% and 38.7%, respectively, which were higher than in female adolescents. Participants who had more than 13 h of sedentary time per day comprised 23%; the percentage was higher among female adolescents (27.4%) than among male adolescents (19.0%). Participants who did not engage in any physical activity for the entire week comprised 61.1%, with the percentage being greater among female adolescents (75.5%) than among male adolescents (47.7%). Regarding weekday sleep duration, 35.8% of the participants slept ≤ 6 h per day during the week, with the percentage being higher among female adolescents (45.2%) than among male adolescents (27.1%). Regarding weekend sleep duration, 55.4% of the participants slept more than 8 h per day on the weekends.

Regarding mental health problems, 28.6% of the participants had high perceived stress, 8.1% had depressive moods, and 3.8% had suicidal thoughts. Notably, these rates were all higher in female adolescents than in male adolescents. The prevalence of depressive mood and suicidal thoughts was more than two-fold higher in female than in male adolescents (Table 1).

### 3.2. Mental Health Problems according to Participants’ Characteristics

Table 2 shows the mental health problems in relation to the participants’ sociodemographic, health status, and lifestyle habits characteristics. The rates of stress perception, depressive mood, and suicidal thoughts were higher among adolescents with poor perceived health (all *p* < 0.001). In terms of BMI, stress perception was highest in the underweight and obese groups (*p* = 0.003).

The rate of depressive mood was high in the smoking group (*p* = 0.016). The rates of stress perception, depressive mood, and suicidal thoughts were higher in the drinking group (*p* ≤ 0.001). The rates of stress perception and depressive mood were the highest in the group who had more than 13 h of sedentary time (*p* < 0.001 and *p* = 0.009, respectively). The group with no physical activity, who did not engage in any physical activity for the entire week, had a high stress perception (*p* = 0.001). In terms of weekday and weekend sleep duration, the group who slept ≤ 6 h had high rates of stress perception, depressive mood, and suicidal thoughts (*p* ≤ 0.001; Table 2).

### 3.3. Factors Associated with Mental Health Problems

To examine the association between mental health problems and lifestyle habits adjusted for sociodemographic and health status characteristics, we used multi-nominal logistic regression analyses calculated using a complex sample analysis (Table 3). The references for the dependent variables were low stress perception, no depressive mood, and no suicidal thoughts. The regression models with sociodemographic, health status, and lifestyle habits factors significantly explained the stress perception, depressive mood, and suicidal thoughts.

Stress perception was low in the 16–18-year-old group and was high among females, those with poor perceived health, and underweight participants. In particular, the odds of high stress perception were higher among those who drink alcohol (odds ratio (OR) = 1.61, 95% confidence interval (CI) 1.19–2.19), those who had longer sedentary time (781 ≤ min/day; OR = 1.88, 95% CI 1.36–2.58), and those with ≤ 6 h of sleep on weekdays (OR = 1.28, 95% CI 1.00–1.66). In contrast, the odds for high stress perception were low among those who slept more than 8 h on weekends (OR = 0.74, 95% CI 0.56–0.98).

Depressive mood was high among females, those with poor perceived health, and underweight participants. The odds for depressive mood were higher among those who drink alcohol (OR = 2.59, 95% CI 1.67–4.02) but low among those who did not engage in physical activity for the entire week (OR = 0.59, 95% CI 0.40–0.88) and who slept more than 8 h on the weekends (OR = 0.63, 95% CI 0.41–0.98).

Suicidal thoughts were high among females but low among overweight participants. The odds for suicidal thoughts were high among those who drink alcohol (OR = 2.28, 95% CI 1.18–4.42) and among those with ≤ 6 h of sleep on weekdays (OR = 1.98, 95% CI 1.05–3.75) but low among those who slept more than 8 h on weekends (OR = 0.41, 95% CI 0.21–0.79; Table 3).

## 4. Discussion

This study aimed to identify the factors significantly associated with mental health problems (stress perception, depressive mood, and suicidal thoughts) in adolescents. The sociodemographic and health status factors of age, sex, poor perceived health, and BMI were identified as predictors of stress perception, depressive mood, and suicidal thoughts. The lifestyle habit factors of drinking alcohol, sedentary time, physical activity, weekday sleep, and weekend sleep were identified as being associated with mental health problems.

In this study, drinking alcohol was commonly associated with increased stress perception, depressive mood, and suicidal thoughts, and adolescents with alcohol drinking experience had a higher risk of mental health problems than adolescents who did not drink alcohol. Adolescent drinking is a public health problem that is known to induce injury and bodily harm, cause acute and chronic health problems, and cause serious negative social outcomes [21,33]. Adolescent drinking was also found to increase the likelihood of short-term and long-term mental health and neurocognitive problems [34,35]. In this research, the drinking group showed higher rates of stress perception, depressive mood, and suicidal thoughts than non-drinkers.

Our results show that 35.4% of adolescents reported that they had drunk more than one cup of alcohol in their lifetime. Alcohol is one of the substances most abused by adolescents globally [36,37,38], thus, it is important to understand the gravity of the problem of adolescent drinking. According to the 2018 National Survey on Drug Use and Health, 19% of adolescents in the United States aged 12–20 years drank alcohol in the past 30 days [37]. As of 2019, the current alcohol consumption rate among Korean adolescents aged 13–18 years (percentage of people who drank at least one drink of alcohol in the past 30 days) was 15% [38]. The WHO has promoted various social measures that have been implemented worldwide to prevent adolescent drinking [21]. Both the Korean government and private organizations provide various drinking prevention projects and services. In Korea, a drinking prevention project aimed at improving adolescents’ health behaviors and health status was launched in school health systems in accordance with the School Health Act and Health Plan 2020 [39,40]. However, because this project only provides a one-time health education—and it is only for students—continuous drinking prevention education that targets students, teachers, parents, families, and communities is also needed.

Sedentary time was significantly associated with stress perception in this study. It was found that an increase in sedentary time was associated with an increase in stress perception in adolescents. Sedentary time has been found to interact with adolescents’ emotional and mental health outcomes [22,23,41]. Screen time for leisure (entertainment or for-leisure use of electronic media, such as television, electronic games, and computers) was strongly associated with adolescents’ depressive symptoms and psychological distress [23], and increased sedentary behavior was found to be associated with depression and decreased psychological well-being [22]. Furthermore, a study of adolescents from five Asian countries also observed that single and multiple types of psychological distress and substance use (e.g., cigarette, alcohol) increased with increasing leisure-time sedentary behaviors [41].

According to our study, 61% of adolescents spend more than 10 h per day sitting or lying down. In particular, the stress perception of adolescents with a sedentary time of 13 h or longer was 1.88 times higher than that of adolescents with a sedentary time of 10 h or less. Today, the importance of public health interventions that attempt to decrease the incidence of depression among adolescents by promoting light activity and decreasing sedentary behaviors is emphasized [42]. Adolescents spend most of their day at school and, because school is recognized as a key environment for encouraging adolescents’ physical activity, interventions to improve physical activity and reduce sedentary time at school are increasing [43]. According to the results of a meta-analysis of interventions to improve physical activity and reduce sedentary behavior in school environments, school recess interventions increase moderate-to-vigorous physical activity (MVPA) [43]. Therefore, physical activity is needed to reduce sedentary time during school hours. The COVID-19 outbreak replaced the attendance of lessons in the classroom with home schooling and online learning activities, so adolescents’ physical activity decreased as time spent inside the house increased [44]. Therefore, a policy to create break times even during online school classes and intentionally reduce sedentary time is needed.

In this research, 61.1% of participants did not engage in any physical activity. Many prior studies have reported that exercise has a positive effect on mental health [22,45] but our results showed that the physical inactivity group had a lower rate of depressive mood. The variable number of days a week with 60 min ≤ physical activity/day was created for adolescents in the 2017 and 2018 KNHANES. It is a simple item that asks about the number of days of physical activity and does not give information about the type, duration, and intensity of physical activity. Thus, subsequent studies should also include these variables in addition to the number of days a week with 60 min ≤ physical activity/day for continued research and validity testing. According to a systematic review and meta-analysis by Rodriguez-Ayllon et al. [22], physical activity is significantly associated with greater levels of psychological well-being. Further, adequate exercise improves social and emotional functions along with cognitive and academic development and relieves depression [45]. A recent 6-year prospective cohort study by Kandola et al. [42] reported that light activity helps lower the risk of depressive symptoms. Kandola et al. [42] argue that, in addition to emphasizing the effects of MVPA such as brisk walking or cycling, it is important to stress the importance of light activity, which requires less effort than MVPA, does not require advance planning, and can easily be incorporated into daily life. In addition, regular physical activity should be encouraged by suggesting that any physical activity is better than none [46].

Sleep duration was consistently associated with stress perception, depressive mood, and suicidal thoughts in this study. Less sleep duration during the week was associated with an increase in stress perception and suicidal thoughts, and long weekend sleep duration was associated with a decrease in stress perception, depressive mood, and suicidal thoughts. Inadequate sleep in adolescents is known to negatively impact their mental health in terms of depression, suicidal thoughts, and increased risk of suicide attempts; hence, improving sleep duration could potentially enhance adolescents’ mental health [24,47]. A study that analyzed the impact of sleep duration on suicidal ideation in Korean adolescents using the 2013 KYRBWS data [25] found a two-fold higher rate of suicide ideation among students who sleep less than 5 h a day. Similarly, in our study, those who sleep less than 6 h a day on weekdays engaged in suicidal thoughts 1.98 times more often than those who sleep 6–8 h a day on weekdays. However, students who sleep more than 8 h a day on weekends showed 0.74 times less stress, 0.63 times less depressive mood, and 0.41 times fewer suicidal thoughts compared to those who sleep 6–8 h a day on weekends, suggesting that adolescents should adequately catch up on sleep on weekends.

In the present study, 35.8% of adolescents reported that they sleep an average of fewer than 6 h per day during the week and only 12.3% of adolescents reported sleeping an average of 8 h per day or more during the week, suggesting a lack of sleep in Korean adolescents. Adequate sleep improves adolescents’ emotional function, memory, and attention, reduces mood lability and impulsivity, has positive effects on school life, such as grades and attendance, and promotes quality of life and physical and mental health [47,48]. According to a study by Foti et al. [49], physical activity of more than 60 min per day and limited sedentary time were associated with sufficient sleep in adolescents. Therefore, public health approaches that consider the potential impact of adequate sleep should be designed and implemented to benefit adolescents. Furthermore, additional studies are needed to examine the causality between adolescents’ sleep duration and mental health problems.

Our study used the newly created physical activity, weekday sleep, and weekend sleep variables from the recent data by KNHANES VII. Our findings were able to identify the common factors significantly associated with lifestyle habits and mental health problems, such as stress perception, depressed mood, and suicidal thoughts in Korean adolescents. In conclusion, alcohol moderation education, timed breaks to reduce sedentary lifestyles, school education with increased physical activity and reduced sedentary time, and education and campaigns to promote adequate sleep duration and catching up on sleep on weekends are needed to reduce mental health problems among adolescents. To increase compliance with lifestyle habits for the promotion of adolescents’ mental health, caregivers, family, teachers, schools, and health care professionals should reinforce their message with stronger arguments and, furthermore, interventions that consider lifestyle habits commonly associated with mental health should be made. Future studies should be conducted to develop these health-promoting activities and assess their effectiveness.

### Study Limitations

This study has a few limitations. First, it is a cross-sectional, secondary data analysis, so there are limitations in interpreting the causality between mental health problems and lifestyle habits. Thus, prospective and longitudinal research designs with more sophisticated high-quality designs and methods should be used to understand the causal and reciprocal relationship between health problems and lifestyle habits. Second, we conducted secondary data analysis using the KNHANES data. This study is vulnerable to recall bias and response bias of self-reported information and to reliability due to under- or over-reporting as the data were taken from the self-reported KNHANES. Furthermore, only a limited number of lifestyle habit variables were available for analysis. Hence, subsequent studies should use primary data that reflect multiple lifestyle habits, and should adopt more comprehensive and valid instruments to measure stress perception, depressive mood, and suicidal thoughts. Third, while we used physical activity (number of days a week with 60 min ≤ physical activity/day) as one of our main variables, it was only collected for adolescents in 2017 and 2018 in the 3-year 7th KNHANES. Therefore, future studies should apply this variable and examine its long-term effects.

## 5. Conclusions

This study examined mental health problems and identified associated lifestyle habits in Korean adolescents using a nationally representative dataset. Adolescents’ drinking alcohol, sedentary time, physical activity, and weekday and weekend sleep were significantly associated with mental health problems. This study focused on adolescents’ lifestyle habits that are modifiable through public health intervention, and the results emphasize the importance of primary care to promote mental health and prevent disease by reducing emergence and exacerbation. Moreover, the findings highlight the need to provide education to improve adolescents’ health behaviors by involving caregivers, family, teachers, schools, and mental health care professionals in the prevention effort.

## Figures and Tables

**Figure 1 ijerph-17-09418-f001:**
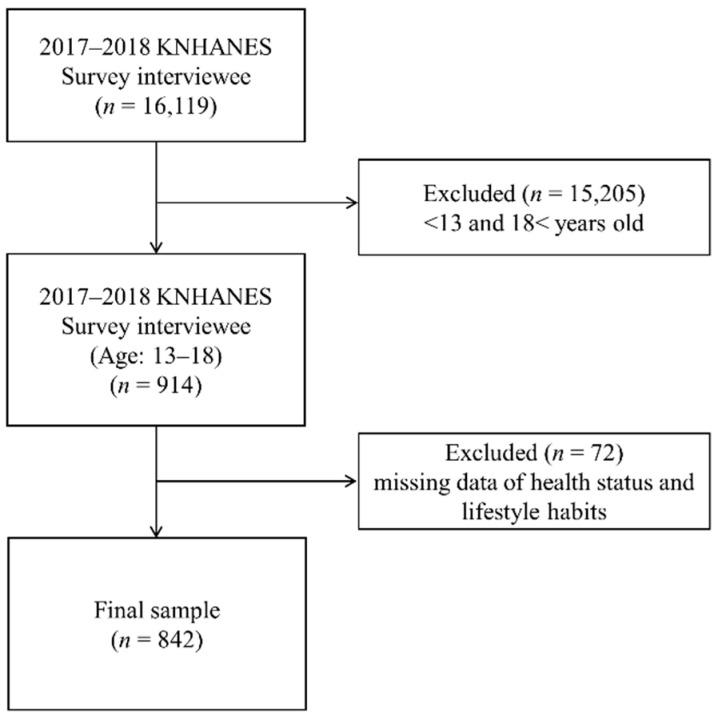
Flowchart of study sample.

**Table 1 ijerph-17-09418-t001:** General characteristics of the participants.

Variables/Categories	Total	Men	Women
Unweighted number	842	440	402
Total (% ^a^)	100.0	51.9	48.1
Sociodemographic characteristics			
Age (years)			
13–15	44.2	44.0	44.4
16–18	55.8	56.0	55.6
Residential area			
Rural	10.9	9.8	12.2
Urban	89.1	90.2	87.8
Household income level			
Low	10.8	10.6	11.0
Middle low	22.7	23.4	21.9
Middle high	32.2	30.8	33.9
High	34.3	35.2	33.2
Health status characteristics			
Perceived health status			
Poor	5.5	5.2	5.8
Fair	36.4	32.3	40.7
Good	58.1	62.5	53.5
BMI percentile			
Underweight	8.0	9.3	6.6
Normal	70.7	69.0	72.5
Overweight	8.3	8.0	8.6
Obesity	13.0	13.7	12.3
Lifestyle habits characteristics			
Smoking			
Yes	14.0	18.7	9.1
No	86.0	81.3	90.9
Drinking alcohol			
Yes	35.4	38.7	31.9
No	64.6	61.3	68.1
Sedentary time (min/day)			
≤600	39.0	41.5	36.2
601–780	38.0	39.5	36.4
781≤	23.0	19.0	27.4
Physical activity			
Yes	38.9	52.3	24.5
No	61.1	47.7	75.5
Sleep duration per weekday (min/day)			
≤360	35.8	27.1	45.2
361–480	51.9	58.2	45.1
481≤	12.3	14.7	9.7
Sleep duration per weekend (min/day)			
≤360	8.4	8.4	8.4
361–480	36.2	37.4	34.9
481≤	55.4	54.2	56.7
Mental health problems			
Stress perception			
High	28.6	24.5	32.9
Low	71.4	75.5	67.1
Depressive mood			
Yes	8.1	5.3	11.2
No	91.9	94.7	88.8
Suicidal thoughts			
Yes	3.8	2.0	5.6
No	96.2	98.0	94.4

Note: BMI = body mass index. ^a^ Weighted percentages calculated by a complex sample analysis.

**Table 2 ijerph-17-09418-t002:** Mental health problems according to participants’ characteristics.

Variables/Categories	Stress Perception		X^2^	*p*	Depressive Mood		X^2^	*p*	Suicidal Thoughts		X^2^	*p*
	High	Low			Yes	No			Yes	No		
Total (% ^a^)	28.6	71.4			8.1	91.9			3.8	96.2		
Sociodemographic characteristics												
Age (years)												
13–15	28.4	71.6	0.12	0.875	6.2	93.8	3.21	0.011	4.0	96.0	0.09	0.709
16–18	28.8	71.2			9.7	90.3			3.6	96.4		
Sex												
Men	24.5	75.5	7.24	<0.001	5.3	94.7	9.78	<0.001	2.0	98.0	7.43	<0.001
Women	32.9	67.1			11.2	88.8			5.6	94.4		
Residential area												
Rural	23.6	76.4	1.26	0.246	10.5	89.5	0.74	0.237	2.6	97.4	0.39	0.366
Urban	29.2	70.8			7.9	92.1			3.9	96.1		
Household income level												
Low	23.7	76.3	2.34	0.261	6.5	93.5	1.93	0.340	5.1	94.9	0.73	0.730
Middle low	30.2	69.8			10.2	89.8			3.2	96.8		
Middle high	26.9	73.1			7.0	93.0			3.5	96.5		
High	30.8	69.2			8.4	91.6			4.0	96.0		
Health status characteristics												
Perceived health status												
Poor	59.6	40.4	37.61	<0.001	30.1	69.9	32.69	<0.001	13.2	86.8	13.73	<0.001
Fair	34.5	65.5			8.4	91.6			4.4	95.6		
Good	21.9	78.1			5.9	94.1			2.5	97.5		
BMI percentile												
Underweight	36.4	63.6	7.50	0.003	13.0	87.0	4.18	0.082	5.6	94.4	1.71	0.296
Normal	26.0	74.0			7.5	92.5			3.6	96.4		
Overweight	30.1	69.9			4.9	95.1			1.8	98.2		
Obesity	36.7	63.3			10.5	89.5			4.7	95.3		
Lifestyle habits characteristics												
Smoking												
Yes	33.6	66.4	1.65	0.077	12.3	87.7	3.14	0.016	5.8	94.2	1.49	0.130
No	27.8	72.2			7.5	92.5			3.4	96.6		
Drinking alcohol												
Yes	34.5	65.5	7.86	<0.001	13.6	86.4	18.01	<0.001	6.2	93.8	7.52	0.001
No	25.4	74.6			5.2	94.8			2.4	97.6		
Sedentary time (min/day)												
≤600	21.8	78.2	17.10	<0.001	6.7	93.3	5.37	0.009	2.1	97.9	3.90	0.056
601–780	29.5	70.5			7.2	92.8			4.8	95.2		
781≤	38.7	61.3			12.1	87.9			4.9	95.1		
Physical activity												
Yes	23.9	76.1	5.70	0.001	8.0	92.0	0.03	0.818	2.7	97.3	1.76	0.114
No	31.6	68.4			8.3	91.7			4.5	95.5		
Sleep duration per weekday (min/day)												
≤360	36.7	63.3	17.47	<0.001	11.7	88.3	8.97	0.001	6.5	93.5	10.15	<0.001
361–480	25.3	74.7			6.8	93.2			1.9	98.1		
481≤	18.1	81.9			3.6	96.4			3.5	96.5		
Sleep duration per weekend (min/day)												
≤360	40.4	59.6	13.09	<0.001	15.0	85.0	10.01	<0.001	8.0	92.0	9.53	<0.001
361–480	33.0	67.0			10.3	89.7			5.4	94.6		
481≤	23.8	76.2			5.7	94.3			2.1	97.9		

Note: The total unweighted sample size is 842. BMI = body mass index. ^a^ Weighted percentages calculated by a complex sample analysis.

**Table 3 ijerph-17-09418-t003:** Multi-nominal logistic regression to compare factors associated with mental health problems ^a^.

Variables/Categories	Stress Perception		Depressive Mood		Suicidal Thoughts	
	OR	95% CI	OR	95% CI	OR	95% CI
Sociodemographic characteristics						
Age (years) (ref. 13–15)						
16–18	0.63 ***	0.49, 0.80	0.99	0.67, 1.45	0.59	0.32, 1.09
Sex (ref. Men)						
Women	1.34 *	1.05, 1.69	3.02 ***	1.82, 5.01	2.99 **	1.60, 5.58
Residential area (ref. Rural)						
Urban	1.26	0.76, 2.08	0.70	0.39, 1.28	1.25	0.55, 2.80
Household income level (ref. Low)						
Middle low	1.51	0.97, 2.37	1.58	0.82, 3.05	0.64	0.29, 1.40
Middle high	1.32	0.81, 2.16	1.01	0.49, 2.06	0.69	0.26, 1.81
High	1.39	0.86, 2.24	1.07	0.53, 2.17	0.72	0.30, 1.72
Health status characteristics						
Perceived health status (ref. Fair)						
Poor	2.24 ***	1.34, 3.76	4.65 ***	2.36, 9.16	2.45	0.92, 6.53
Good	0.58 **	0.46, 0.74	0.77	0.49, 1.21	0.63	0.34, 1.19
BMI percentile (ref. Normal)						
Underweight	1.52 *	1.05, 2.21	1.91 *	1.07, 3.42	1.32	0.51, 3.42
Overweight	1.14	0.77, 1.70	0.58	0.24, 1.40	0.38 ***	0.23, 0.61
Obesity	1.22	0.81, 1.83	1.02	0.58, 1.79	0.89	0.44, 1.81
Lifestyle habits characteristics						
Smoking (ref. No)						
Yes	1.19	0.83, 1.70	1.49	0.84, 2.63	1.72	0.75, 3.96
Drinking alcohol (ref. No)						
Yes	1.61 **	1.19, 2.19	2.59 ***	1.67, 4.02	2.28 *	1.18, 4.42
Sedentary time (min/day) (ref. ≤600)						
601–780	1.35 *	1.02, 1.78	0.90	0.55, 1.49	2.03	0.84, 4.86
781≤	1.88 ***	1.36, 2.58	1.42	0.81, 2.50	1.66	0.72, 3.81
Physical activity (ref. Yes)						
No	1.21	0.95, 1.55	0.59 *	0.40, 0.88	1.00	0.50, 2.00
Sleep duration per weekday (min/day) (ref. 361–480)						
≤360	1.28 *	1.00, 1.66	1.01	0.65, 1.56	1.98 *	1.05, 3.75
481≤	0.69	0.45, 1.05	0.61	0.24, 1.55	2.30	0.89, 5.99
Sleep duration per weekend (min/day) (ref. 361–480)						
≤360	1.23	0.74, 2.04	1.81	0.89, 3.67	1.40	0.61, 3.20
481≤	0.74 *	0.56, 0.98	0.63 *	0.41, 0.98	0.41 **	0.21, 0.79
Cox and Snell R^2^	0.09		0.08		0.05	
Nagelkerke R^2^	0.13		0.18		0.17	

Note: The total unweighted sample size is 842. BMI = body mass index; CI = confidence interval; OR = odds ratio. ^a^ Multi-nominal logistic regression calculated by a complex sample analysis. * *p* < 0.05, ** *p* < 0.01, *** *p* < 0.001.

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
