# Peer review of "Factors Associated with Lifestyle Habits and Mental Health Problems in Korean Adolescents: The Korea National Health and Nutrition Examination Survey 2017–2018"

_ijerph, 2020, doi:10.3390/ijerph17249418_

Round 1
Reviewer 1 Report
See attached comments.
In “What are the Modifiable Health Risk Behaviors associated with Korean Adolescents’ Mental Health?” the authors explore the relationships between a variety of health factors and mental health problems (i.e., stress, depressive mood, and suicidal thoughts) in Korean adolescents through a cross-sectional design.
Broadly, I think that the motivation for these analyses is not completely clear. A quick search on Google Scholar or PubMed brings up a number of articles examining "correlates" and mental health variables in this age group. I think a case can be made that more research is needed on mental health, but not sure that case is actually made in the manuscript’s current form, or if the analyses are addressing literature gaps/limits. Other papers seem to look at these outcomes separately, but also include a number of independent variables in the models. For example, the authors mention that a holistic approach is needed. Does that mean that previous papers did not include all relevant independent variables in the same model? The independent variables were not motivated and/or explained. The introduction could benefit from more information on these health factors of interest. Relatedly, there I had concern with the Measurements section of the Methods. The measures should have supporting reliability/validity information for these variables and justification/explanation for categories that were used. In general, the Discussion section could use some clarity (e.g., topic sentences for a few paragraphs to understand the direction/presentation of information).
Thank you very much for your careful review and another opportunity to submit the revised manuscript. We very much appreciate the thoughtful comments and feel that the paper has been greatly improved. Thank you, your comment was very helpful. The difference between previous research and our research is described in the Introduction and Discussion. In addition, the reasons for choosing the independent variables and the necessity of our study have been rewritten in the Introduction.
We have added the description of variables and reliability/validity information for variables in Materials and Method section according to your comment.
In order to clarify the Discussion section, we have added the topic sentence of the paragraph, and we have revised the Discussion section according to your comment.
Abstract
Line 22: Agreed - but physical activity was not one of the listed variables - sedentary time was.
Thank you for your comment. We have deleted the word based on your comment
Introduction
Line 28: "correct" assumes habits are already poor. Is this the case?
Thank you for your comment. We have changed "correct" to “develop”.
Line 30: These two sentences seem to have a different focus. Talking about long-term influence of both, but the first sentence talks about health/behaviors and then it seems to jump to mental health problems. This paragraph feels a little disjointed and incomplete.
Thank you for your comment. We agree with your opinion. We have revised the first paragraph of the Introduction section accordingly.
Line 34: How are mental health problems being defined in this paper? Later we learn that the authors are referring to stress, depression, suicidal thoughts - make that clear early on.
Thank you for your comment. In this paper, mental health conditions were defined as mood disorders, anxiety disorders, impulse control disorder, and substance use disorders. We have revised the first paragraph of the Introduction section based on your comment.
Line 44-45: Not sure what is meant here.
Thank you. We agree with your opinion. We have corrected this sentence to be accurate and concise based on your comment.
The modified sentence is as follows: Stress, depression, and suicide are on the mental health continuum and are correlated.
Line 70: But, what are the authors considering here as "mental health problems"? Stress, depression, and suicidal thoughts were introduced in previous paragraph as examples - are they the outcomes of this paper? There are other potential variables that could fit this category of mental health problems, so why these?
We very much appreciate the thoughtful comments and feel that the paper has been greatly improved. We agree with your opinion.
Among the Organization for Economic Cooperation and Development countries, the prevalence of depression and suicide among adolescents in Korea is relatively high and suicide is the number one cause of death in Korean adolescents. Therefore, Korea considers adolescents' stress, depression, and suicide as continuation of mental health problems, so they are considered an important public health problem and are investigated every year through national surveys. As adolescents’ mental health problems are clinically heterogeneous and the interaction of different health risk factors increases the risk of mental health problems, it is difficult to understand the etiology of mental health problems in adolescents. Previous studies examined the association between various lifestyle habits and mental health problems in adolescents. However, there is limited information reflecting the average time of sedentary time and weekdays and weekends sleep in Korean adolescents using large-scale national data from an anonymous survey. In addition, there are limited studies that examine the association between mental health problems and sedentary time and sleep lifestyle habits in Korean adolescents.
We have revised the Introduction section based on your comment.
Line 70: This sentence is not clear to me. By mentioning "onset" it makes one assume that a longitudinal approach will be taken, but this is not the case?
Thank you. We agree with your opinion. We have corrected this sentence to be accurate and concise based on your comment.
The modified sentence is as follows: When considering the possibility of co-occurrence of various health risk behaviors, it is necessary to consider various modifiable lifestyle habits to understand and manage the mental health problems of adolescents.
Line 71: What is meant by holistic approach?
Thank you. We have corrected the ambiguity of the sentence based on your comment.
The modified sentence is as follows: When considering the possibility of co-occurrence of various health risk behaviors, it is necessary to consider various modifiable lifestyle habits to understand and manage the mental health problems of adolescents.
Line 74: I think there needs to be some motivation of why these variables were examined. What is already known or not know about these behaviors and the outcomes?
We very much appreciate the thoughtful comments and feel that the paper has been greatly improved. We agree with your opinion.
There is limited information reflecting the average time of sedentary time and weekdays and weekends sleep in Korean adolescents using large-scale national data from an anonymous survey. In addition, there are limited studies that examine the association between mental health problems and sedentary time and sleep lifestyle habits in Korean adolescents. Therefore, this study uses the variables of physical activity, sedentary time, and weekdays and weekends sleep duration from the nationally representative sample of Korean adolescents recently generated by the 7th Korea National Health and Nutrition Examination Survey (KNHANES). In addition, this study understands the recent trends in lifestyle habits and mental health in the Korean adolescent population to enable the development of preventive measures to address modifiable lifestyle habits.
We have revised the Introduction section based on your comment.
Line 80: Currently, I am not sure what this is adding to the literature. Why do this versus a review? Why is new data needed? And have previous studies been cross-sectional or longitudinal. Seeing below that this is cross-sectional and we cannot explore temporal association, why is this needed if each problem has been studied before? Is there a need to examine this in a Korean sample to see how findings compare to other studies? This could be justified, but currently it is not clear.
Thank you. We have justified the need for our research throughout the Introduction section based on your comment.
Line 82: Has this not been done? It makes sense to do, but not motivated in the intro that this analysis is needed/not known.
Thank you. We have justified the need for our research throughout the Introduction section based on your comment.
Line 111: Make plural
Thank you for your comment. We have revised the word accordingly.
Lines 112-121: Any reliability/validity info to support/justify these questions/measures or the KNHANES as a whole?
Thank you. We have added the sentence based on your comment.
KNHANES, which started in 1998, is a nationwide survey that calculates national statistics through surveys on health level, health-related consciousness and behavior, and food and nutrition intake of 10,000 people every year.
The contents of the KNHANES are selected by seeking the opinions of the central government and experts in related fields to calculate the statistics necessary to establish and evaluate health policies.
Each year, the demand for indicators of the Ministry of Health and Welfare is checked. The needs of the survey contents, survey items, survey methods, calculation indicators, and results use are reviewed by the subcommittee advisory committee and the coordination advisory committee and, then, reflected upon during the following year.
The questions on stress perception, depressive mood, and suicidal thoughts were used as dependent variables in our study. These questions are the same that have been used since 1998. Statistics on health levels have been produced and various health policies have been promoted based on the results of studies that have used these variables.
Line 129: Were 13-year olds aware of household income? It seems maybe a census type source was used but could this sentence be worded with a little more clarity
Thank you. Household income levels were surveyed on one adult per household (age 19 and over), and we have added this based on your comments.
Line 147: Why were these thresholds chosen?
Thank. We have added the sentence based on your comment.
The average sedentary time differs between countries and, in Korea, it also differs for adolescents. Therefore, in this study, we used sedentary minutes divided into quartiles from representative data of KNHANES. The result is as follows.
≤ 600 (≤ 25th percentile): 39%
601–660 (25th < percentile ≤ 50th percentile): 8.5
661-780 (50th < percentile ≤ 75th percentile): 29.5
781 ≤ (75th < percentile): 23%
We classified the sedentary time into three groups: ≤ 600 (≤ 25th percentile), 601–780 (25th < percentile ≤ 75th percentile), and 781 ≤ (75th < percentile; min/day).
Line 152: Again, why were these categories selected?
Thank you. We have added the sentence based on your comment.
The physical activity variable we used in this study was a variable used for the first time in KNHANES VII–2 and 3 (2017–2018) for adolescents aged 12-18 years. Physical activity refers to the number of days a participant engaged in physical activity for 60 minutes or more per day in a week. Before the variables are categorized, the results of the descriptive statistics confirmed are as follows.
None: 61.1%
1 day/week: 8.4%
2 days/week: 11.7%
3 days/week: 8.1%
4 days/week: 2.9%
5 days/week: 3.8%
6 days/week: 1.7%
7 days/week: 2.3%
We tried to compare the group who did exercise with the group who did not exercise, as 61.1 percent of the non-exercise groups accounted for a large portion. Therefore, as we focused on whether adolescents engaged in physical activity or not, physical activity was classified as yes (1, 2, 3, 4, 5, 6, or 7 days/week) and no (no days/week).
If we need to re-establish the classification criteria, we will do analysis again. Thank you.
Line 153: How was sleep assessed? What was/were the questions?
Thank you for your comment. Weekday sleep and weekend sleep refer to the average time spent sleeping per day, which was calculated in minutes using the time between when the participant went to bed and when the participant woke up. We have added the sentence to the sleep assessment method and question based on your comment.
Line 155: Again, justification for these categories?
Thank you for your comment. According to the recommendations of the Korean Society of Sleep Medicine, it is generally recommended to sleep for 6 to 8 hours a day to be healthy. We have added the sentence based on your comment.
Results Line 185: I recommend referring to Table 1 in text sooner.
Thank you for your comment. We have referred to Table 1, Table 2, Table 3 sooner in the text.
Discussion Line 265: I suggest starting with a topic sentence to understand the focus of this paragraph. Starting off with a statement about a country not assessed in this study is confusing. It is hard to follow the message in this paragraph
Thank you for your comment. We have revised this paragraph to clarify this statement. The topic was presented at the beginning of the paragraph.
Line 266: US?
Thank you for your comment. The 2018 National Survey of Drug Use and Health data represents results from adolescents in the United States. Therefore, we have added "in the United States" in the text.
Line 283: The first part of this paragraph belongs in the intro to justify some of the independent variables that were examined
Thank you. We agree with your opinion. We have deleted this paragraph from the Discussion and added this paragraph to the Introduction section based on your comment.
Line 293: I would imagine that this longitudinal study included covariates in the model - did they not?
This is the type of literature reference that I think was missing in the intro, but it still doesn't help highlight why the current study looked at these questions, and with a cross-sectional study.
Thank you. We agree with your opinion. We have deleted this paragraph as the explanation of this longitudinal study did not seem to help to highlight the reason for conducting this study
Lines 331-334: Agreed. But, I believe this message/conclusion has already been made from previous studies.
Thank you. We have revised this paragraph based on your comment.
Line 338: But there have been some of these. Were there concerns/limitations with previous prospective studies?
Thank you. We have looked for limitations in previous prospective studies and we have revised the limitation based on your comment.
Line 340: I would argue that misclassification of both exposure and outcome could be significant concerns with the measurements of these variables. Any thoughts on generalizability?
Thank you for your comment. We have revised the limitation accordingly.
In terms of generalization, this study was conducted by selecting participants representative of Korean adolescents using the stratified cluster sampling method. Therefore, the study can be generalized using the weighted sample of complex design. Generalization was also mentioned in the Materials and Methods section (“The findings of these study could be generalized to Korean adolescents”) so I did not describe it again in Limitation section, but if it is necessary, I will.

Author Response
"Please see the attachment."

Reviewer 2 Report
This is an interesting, easy to understand paper with well presented results. The authors have tried their best to put results and discuss them in simple language. My suggestions/ comments for paper entitled "What are the modifiable health risk behaviors associated with Korean adolescents' mental health?" are as follows:
Thank you very much for your careful review and the opportunity to submit the revised manuscript. We appreciate the helpful comments and feel that the paper has been greatly improved.
Please add some more information on health risk behaviors (sleep, sedentary lifestyle) and its association with mental health in the background.
Thank you for your comment. We have added more information about sleep and sedentary lifestyle.
Please add hypothesis set for the study conducted.
Thank you, we have added hypothesis based on your comment.
As this study has been conducted using the data from KNHANES, please explain the novelty of the study in the background.
Thank you for your comment. KNHANES comprises valid and reliable data representative of the health level and health behavior of the population. In the Introduction, we described the novelty of using KNHANES data.
Author Response
"Please see the attachment."
Reviewer 3 Report
The focus on this paper is the association between health risk behaviors and mental health problems in adolescents. Data from a sample of 842 participants, proceeding from a national survey were extracted. Concretely, background dimensions and health risk behaviors were considered as possible predictors of stress, depression, and suicidal thoughts.
The research question is relevant, although arguments to highlight its relevance should be improved, both with respect to psychological processes implied and the structure of language expressions. On the other hand, more details on the method and analysis should be given.
The title does not fit well with the content of the text. The modifiability of risk behaviors is not treated in the manuscript, and the relevance of specific risk behaviors and mental health indicators selected for the study are not justified.
With respect to Results section, a lot of information in the text overlaps with that given in the tables.
In the Discussion a deeper approach is recommended. Beyond comparing findings of the present study with those of previous investigations, at least tentative explanations should be given with respect to the relations observed between independent and dependent variables.
Thank you very much for your careful review and the opportunity to submit the revised manuscript. We appreciate the thoughtful comments and feel that the paper has been greatly improved.
Thank you, your comment was very helpful. We have revised the title and overall part of the manuscript in detail accordingly. As your comment indicates, the manuscript does not address the possibility of modifying risky behavior.
Therefore, the “health risk behavior” was modified to “lifestyle habit.”
The difference between previous research and our research is described in the Introduction and Discussion. In addition, the reasons for choosing the independent and dependent variables and the necessity of our study have been rewritten in the Introduction. We have revised the Materials and Methods section of manuscript in detail based on your comment. In addition, we have reduced the text of the Results section based on your comment.
The following specific suggestions should be considered:
In the Introduction, a lot of information on the results is given that could be summarized, leaving some space to antecedents of the issue.
Thank you for your comment. We have deleted the unnecessary content and revised the Introduction section to more accurately indicate the need for research.
Sentence connectors should be reviewed along the text. For example:
- See the connector “In particular” in the second sentence of the Introduction. The idea expressed in that sentence is not a concretion of that asserted in the first sentence.
- See the connector “However” in line 62. The idea expressed afterwards does not contradict that asserted in the previous sentence.
Thank you for your comment.
- We have deleted the “In particular”.
- We have deleted the “However”.
Notes in the tables should be reviewed.
- Some information given in them could be best placed in the text, such as statistical analysis details (treatment for missing data, percentile calculation…).
- Replace “:” with =.
Thank you for your comment.
- We have deleted the notes of the tables and added in the text based on your comment. In addition, we deleted the BMI percentile content of the table notes that duplicate the text.
- We have changed “:” to “=”.
Bolds in the body of the tables are not in accordance with APA style.
Thank you, we have deleted the bold text in the body of tables based on your comment.
The post-hoc analysis applied to examine groups’ differences should be indicated.
Sorry, I did not understand your question. The dependent variables in our study were stress perception, depressive mood, and suicidal thoughts but each variable was classified as Yes or No. Therefore, it is not clear to which table the posthoc analysis corresponds. If you can let us know, I will run the statistical analysis again.
In the Discussion, the aim of the study is not correctly expressed. “Indexes of” instead of “factors associated with” seems more appropriate.
Thank you for your comment. The aim of this study was to examine the factors significantly associated with lifestyle habits and mental health problems in adolescents. We have selected words “factors associated with” commonly used in the description at the beginning of the manuscript. Thus we have kept the word the same. However, if it is necessary, I will.
The following reference is recommended regarding social outcomes of adolescent drinking, that are mentioned in line 272 of de Discussion:
Tinajero, C., Cadaveira, F., Rodríguez, M. S., & Páramo, M. F. (2019). Perceived social support from significant others among binge drinking and polyconsuming Spanish university students. International Journal of Environmental Research and Public Health, 16(22), 4506.
Thank you for your comment, we have added the reference. The reference you gave us was very helpful.
Some minor issues:
The terms “Table” and “Figure” in the text should not be in bold.
Thank you for your comment. We have revised it accordingly.
“Dependent Variable” in line 111 should be in plural.
Thank you for your comment. We have revised the word accordingly.
In the expression “p value/s” the term “value/s” should be omitted.
Thank you for your comment. We have deleted all the “value/s” terms from the manuscript based on your comment
Detail on the sample size should not be included in the tittle of the tables.
Thank you for your comment. We have deleted the sample size in the title of the tables and added the word in the notes of the tables.
Author Response
"Please see the attachment."

Round 2
Reviewer 1 Report
Broad Comments
Thank you for the opportunity to review the revised manuscript, “Factors Associated with Lifestyle Habits and Mental Health Problems in Korean Adolescents: The Korea National Health and Nutrition Examination Survey.” I appreciate the attention and responses that the authors have taken to the reviewer comments. I think the revisions have improved the motivation immensely and have only a few remaining comments/suggestions. In general, if available, reliability information is still needed for the specific measures (or survey as a whole) – rather than simply stating that the survey is valid and reliable. Although the writing is improved, there are some instances where nouns should be switched to plural or singular and some of the new sentences may need some clarifying words/attention. (I noted a few in the specific comments below, but did not note all of them.)
Thank you very much for your careful review and another opportunity to submit the revised manuscript. We very much appreciate the thoughtful comments and feel that the paper has been greatly improved.
1. We have added information about the validity and reliability of the Korea National Health and Nutrition Examination Survey (KNHANES) in response to your comments.
2. In addition, the manuscript has been edited by a professional editing service. We have identified cases where nouns needed to be converted into plurals or singulars, and the entire manuscript has been checked for clarity.
Specific Comments
Introduction
- Line 79: I recommend including a general consensus of what other studies have reported here.
Thank you for your comment. Based on your suggestion, we have added the general consensus of what other studies have reported. In particular, we have added information related to the variables used in our study.
- Line 80: Time is redundant in this sentence. This new sentence is a little unclear and I’m not sure how it differs from the following sentence.
Thank you for your comment. We have checked the meaning of the sentence according to your comment and deleted the duplicate content
- Line 88: I think weekdays/weekdays should be singular.
Thank you for your comment. We have checked this phrase throughout the entire manuscript and corrected it based on your comment.
- Line 90: Maybe “explores” rather than “understands”
Thank you for your comment. We have changed “understands” to “explores” based on your comment.
- Line 92: May “allow” who?
Thank you for your comment. We meant “mental health care professionals.” We have added a sentence based on your comment.
- Line 99: I’m not sure that drinking/smoking behaviors were introduced or highlighted previously in the introduction?
Thank you for your comment. A number of studies have shown that smoking and drinking in adolescence increases the risk of mental health problems in the short and long term. Therefore, national data (World Health Organization [WHO] data) were added as a reference.
- Line 103: I think this hypothesis could use some clarity. As written, it is not clear what is being hypothesized. I assumed the authors mean behaviors will differ in categories of the outcomes?
Thank you for your comment. We have clarified the hypothesis based on your suggestion.
Materials and Methods
- Line 108: Change “uses” to “used”
Thank you for your comment. We have changed “uses” to “used” based on your comment.
- Line 114: If the measures are “valid and reliable”, then statistics or evidence should be provided for each of the specific behaviors
Thank you for your comment.
KNHANES data are widely used by governmental organizations and researchers. The Korean Government has periodically revised national health plans and recently established the National Health Plan 2020 (HP 2020). KNHANES provides health statistics on more than half of the target indicators for HP 2020 goals. In addition, the Korean Government uses KNHANES data and provides comparable health statistics. The Korea Centers for Disease Control and Prevention (KCDC) and related academic societies have managed external quality control programs for all steps (including survey administration, data collection, laboratory analysis, and data processing) as well as internal quality assurance and control procedures.
We were able to check the literature on KNHANES’ validity and reliability related to food security measures and the food frequency questionnaire. However, KNHANES does not provide statistics or evidence for each specific variable item used in this study. Although it is possible to refer to many previous studies that use the KNHANES data, the validity and reliability for each of the variables are not specifically cited in the literature.
For this reason, we have not been able to present the validity and reliability of each variable, but the cited study supports the fact that the entire KNHANES survey is considered valid and reliable national data. We have corrected the sentence citing the following reference.
Reference: Kweon, S.; Kim, Y.; Jang, M.J.; Kim, Y.; Kim, K.; Choi, S.; Chun, C.; Khang, Y.H.; Oh, K. Data resource profile: the Korea National Health and Nutrition Examination Survey (KNHANES). Int. J. Epidemiol. 2014, 43(1), 69–77.
- Line 123: Change “home” to “homes”
Thank you for your comment. We have changed “home” to “homes.”
- Line 187: Was any information asked about naps? (If some Korean adolescents do participate in occasional napping?) If not, specify that only nighttime sleep was assessed.
Unfortunately, KNHANES did not measure naps. The sleep-related variables used in this study assessed only nighttime sleep without taking into account adolescents’ naps.
Therefore, in the Materials and Methods section, we have specified that this variable refers to nighttime sleep
Discussion
- Line 192 and elsewhere: The phrase “in our study” was repeated a bit in the discussion – suggest altering sentences to minimize redundancy.
Thank you for your comment. We have changed the sentences throughout the manuscript to minimize duplication of the phrase “in our study” based on your comment.
- Line 296: I would be cautious with wording as “problematic drinking” was not assessed in the current study with how alcohol use was parameterized.
We agree with your opinion. Since our study does not assess binge drinking, we have removed the phrase that refers to problematic drinking
- Line 349: Rather than “commonly”, I suggest “consistently” or maybe “universally”
Thank you for your comment. We have changed “commonly” to “consistently.”
